# Novel Apex Connection for Light Wood Frame Panelized Roof

**DOI:** 10.3390/ma15217457

**Published:** 2022-10-24

**Authors:** Md Saiful Islam, Ying Hei Chui, Zengtao Chen

**Affiliations:** 1Department of Civil and Environmental Engineering, University of Alberta, Edmonton, AB T6G 1H9, Canada; 2Department of Mechanical Engineering, University of Alberta, Edmonton, AB T6G 1H9, Canada

**Keywords:** apex connection, folding mechanism, FEA, panelized light frame roof

## Abstract

Panelized fabrication of light-frame wood buildings has higher productivity than the traditional stick-built method. However, the roof production process is not very efficient due to the structural system and construction method. This study proposes a novel apex connection that allows for a folding mechanism in a panelized light wood frame roof system. Proof of concept of the proposed connection assembly is presented by a 3D printout of the developed connection. Following the steel design code and timber code, the initial estimation of different parameters, such as the pinhole diameter and number screws, were established. A detailed finite element analysis (FEA) was performed to determine the connection strength requirement for different load case scenarios. The results of the FEA and 3D printout of the assembly show that the proposed connection can provide the required folding mechanism before roof installation and can withstand the load in the unfolding state at service.

## 1. Introduction

The majority of residential buildings constructed in North America are light-frame wood buildings (approximately 90%), mostly in the form of single detached family houses and low-rise multi-story apartments [1,2]. In light wood frame construction, the primary framing material is dimension lumber, which is often utilized in combination with other wood products such as plywood, I-beams, and oriented strand board (OSB) to fabricate a building [3,4,5]. However, in recent decades, application of engineered lumber such as laminated strand lumber (LSL) has increased due to the dimensional stability of this structural composite lumber product and the adoption of an off-site construction process. For example, an Alberta-based prefab company in Canada uses LSL and OSB to produce light frame walls and their wall production is fully automated, whereas floor production uses wood I-joist in combination with a semiautomated process [6]. This type of light wood frame construction is termed panelized construction. Panelized construction of light wood frame homes is drawing attention in North America due to its design flexibility and on-site assembly cost savings [7,8]. It utilizes manufacturing principles to build light-frame wood buildings. This off-site construction process subdivides a building model into subassemblies, such as wall panels, floor panels, and volumetric roof elements, which are manufactured in a factory environment and then shipped to the site for installation. A light-frame panelized-building production facility typically encompasses several workstations such as wall and floor production lines.

As most construction activities in the panelized construction of light wood frame buildings are performed in a manufacturing environment, it is critical to obtain optimal productivity in the production lines [7,8]. In the current panelized construction process, the roof is built using closely spaced wood trusses that support OSB sheathing and roofing materials (on the upper chord) and drywall ceiling materials (on the lower chord). The entire roof of a single detached home is subdivided into four or five small volumetric units based on the floor area and is manufactured on the roof production line [6]. The fabrication of roofs follows the same methodology as stick-built construction. For example, the roof trusses are laid out on a setup jig platform (Figure 1a) in the offsite facility according to the building plan, as shown in Figure 1b. Then, other roof components are added (Figure 1c) to manufacture the small roof modules. All the activities in the roof module production are manual and labour intensive. Consequently, current roof production is not as efficient as other building components, such as the wall or floor. Moreover, transporting the roof volumetric units requires a relatively large number of trailers (to be specific, in the case of an Alberta-based home manufacturer, four trailers are required to transport a 1600 sq ft single-family home) and on-site loading and unloading of trusses increases the overall work duration [9]. Therefore, to improve the current roof construction a holistic approach was developed by Islam et al. [10,11]. In this holistic approach, a gable roof was divided into several sub-elements. The dimensions of these sub-elements were aligned with the production line constraints of offsite facility, transportation trailer capacity, crane lifting limitations, and on-site installation considerations. The complete the panelized roof system for a typical two-storey house with a gable roof comprises the following components (Figure 2): (a) Roof panels, (b) support wall panels, (c) celling frames, (d) beams spanning over two support walls, (e) gable ends, and (f) inter-component connections, including the inclined roof panel-to-support wall, ceiling frame-to-load-bearing shear wall, apex connection, and the support wall-to-ceiling frame.

Due to panelization, major components, such as roof panels, ceiling frames and support walls, can be produced in the automated and semi-automated production line of an offsite facility [10]. For instance, the roof panels (panel-A and panel-B in Figure 2) and support walls are produced using LSL and OSB in the wall production line. In contrast, ceiling frames are built using wood I-joist and an LSL rim board. Consequently, major roof component fabrication is expected to require less production time in contrast to the current process due to the utilization of the current wall and floor panel assembly lines. Moreover, transporting the panelized roof requires only one trailer trip in contrast to the current roof system for the same home size [12]. However, in the panelized roof system, all the components are assembled at the site. Consequently, onsite workload increases significantly [9]. Thus, installation of the inter-component connections must be easy to minimize the on-site workload. This paper presents a concept of a novel apex connection that allows assembling two roof panels (Panel-A in Figure 2) at the offsite facility and folding of roof panels while transporting and the self-locking mechanism facilitates easy lifting and installation of the two connecting panels (Figure 3).

## 2. Novel Triangular Hinge Apex Connection

The main limitation of the panelized roof system is the increased crane lifting number while installing the roof at the site. To reduce the onsite workload, a novel connection mechanism is developed so that multiple panels can be lifted at once. The apex connection for a panelized roof with an 8/12 slope can be used to connect the two Panels (in this case panel-A as shown in Figure 2). The advantage of this connection is that it is self-locking and foldable. The apex connection can be installed at the offsite facility and thus two panels will form a triangle module that can be folded, as shown in Figure 3. This folded state of the panel facilitates easy transportation to the site and a single crane lift is required for installation. Since the connection facilitates a self-locking mechanism, the roof panel installation requires no additional job once the module reaches the proper roof angle. Thereby, this system is expected to reduce a significant amount of the workload at the site. To demonstrate the folding and locking mechanism of the assembly, a full-scale 3D printout using PLA prototyping material was developed. Figure 4 illustrates the folding and unfolding state of the 3D full-scale printout. A video of the folding mechanism can be found in the Appendix A of this paper. The primary components of the connection are shown in Figure 5 and consist of the following 8 parts:Secondary bars to connect the panel rafterPrimary Folding link barsMain lock channelMiddle barSecondary lock channelSide-barsSecondary Folding link barsPins

**Figure 4 materials-15-07457-f004:**
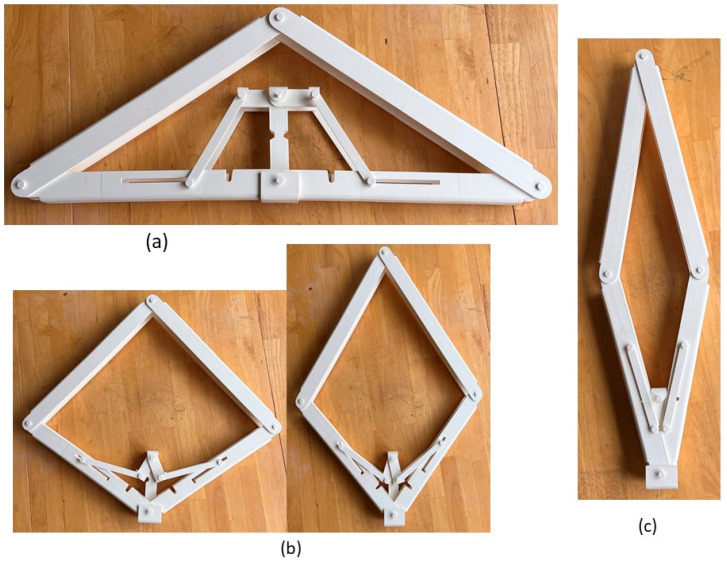
3D printout of full-scale assembly using PLA prototyping material: (**a**) Unfolded state, (**b**) partial folding state, and (**c**) full folding state.

**Figure 5 materials-15-07457-f005:**
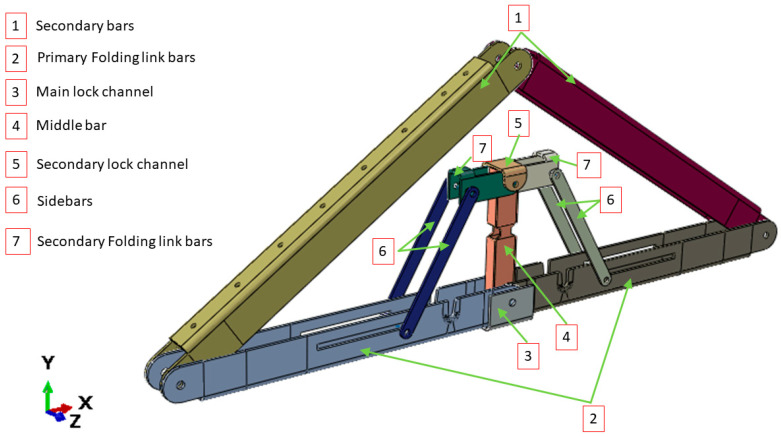
Components of the novel apex connection system for a light frame panelized roof.

The secondary bars are connected to the roof panel rafter using screws, while the primary folding link bars act as rigid link elements when the connection assembly is completely unfolded at the service condition. The primary folding link bar itself has two bars connected using a pin whereas the two secondary bars are tied using pins to the ends of the folding link bars and with each other at the apex point. In order to act as a rigid link, the folding bar has to resist clockwise and counterclockwise rotation depending on the loading condition. The self-locking mechanism is provided through the main lock channel, middle bar, secondary link bars, secondary lock channel and sidebars. It can be observed from Figure 6 that clockwise rotation at point A is resisted by the main lock channel placed at the bottom of the primary folding link bar, whereas counterclockwise rotation is resisted by the secondary lock channel.

All the components can be fabricated using steel sheet metal (12 gauge and 11 gauge). EN 1993-1-8 [13] provides a general guideline for the edge and end distance of pin holes. Figure 7 shows geometrical requirements for pin-ended members according to EN 1993-1-8. The initial estimate of the hole diameter of the assembly was designed considering the steel design guideline by EN 1993-1-8 [13]. For an 11-gauge plate, (yield strength of plate f_y_ = 187 MPa, hole diameter d_o_ = 7.35 mm and design pin force F_ED_ = 8000 N) member the corresponding edge distance and end distance are 10.7 mm and 12.3 mm, respectively. Hole dimensions of the main link bar are governed by the pin diameter requirement, which primarily depends on the shear capacity of the pin. To analyze the connection, a reasonable diameter of the pin was obtained using the pin connection shear capacity equation of EN 1993-1-8 [13]. Following the design guideline and assuming a design value of pin joint dimensions of the parts of apex connection were determined for numerical analysis. Detailed dimensions of all the components of the novel apex connection for a case study roof slope 8/12 are illustrated in Figure 8, Figure 9, Figure 10, Figure 11, Figure 12, Figure 13 and Figure 14.

## 3. Analysis of the Case Study Connection

As can be observed from Figure 3, the apex connection is installed to connect two panels and the main folding bar of the proposed connection at the unfolding state must resist the load at peak of the assembly. This system can be idealized as a statically indeterminate rafter system that has eave support with a collar strut (Figure 15). Using Castigliano’s theorem on deflections with enforcement of displacement compatibility at the redundant reaction, unknown forces of the free body diagram shown in Figure 16 can be obtained using Equations (1)–(4) [14].
(1)$$A_{x}=\frac{wL}{8tan\theta }\left[\frac{-\alpha^{2}+5\alpha +1}{\alpha }\right],$$
(2)$$B_{x}=\frac{wL}{8tan\theta }\left[\frac{\alpha^{2}-5\alpha +3}{\left(1-\alpha \right)}\right],$$
(3)$$C_{x}=\frac{wL}{8tan\theta }\left[\frac{-\alpha^{2}+\alpha +1}{\alpha \left(1-\alpha \right)}\right],$$
(4)$$A_{y}=wL,$$
where:
α = a span factor used to express the location of the interior support or attachment point of the folding link bar*L* = horizontal projection of the distance from the eave to the Apex (the distance in the plan).θ= roof pitch (rafter slope) relative to the horizontal.$x_{1}\;\mathrm{and}\;x_{2}=$ span coordinates measured horizontally, as indicated in Figure 16w= gravity load including the self-weight of the roof panel expressed as a uniformly distributed load

**Figure 15 materials-15-07457-f015:**
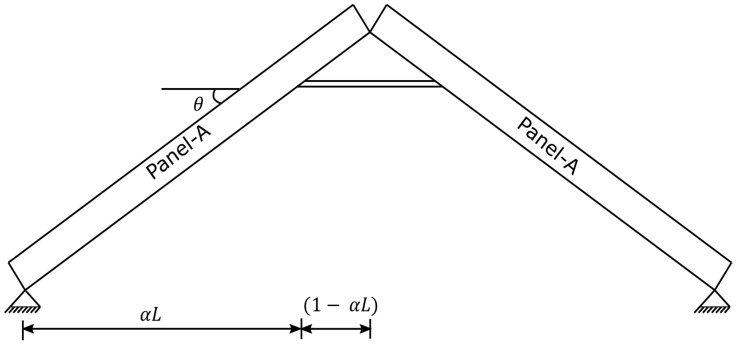
The apex connection idealized as rafter support system eave support with Collar Strut.

**Figure 16 materials-15-07457-f016:**
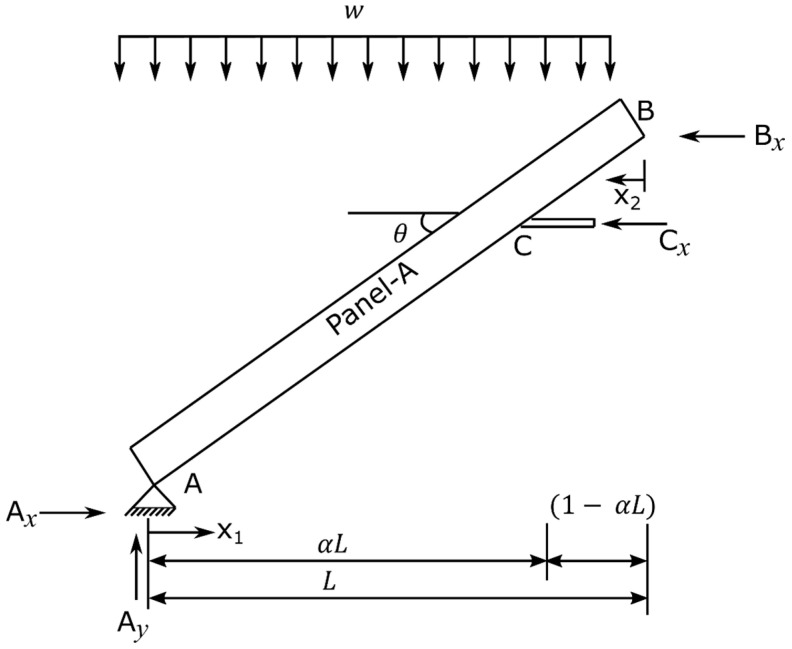
Free body diagram of the system under gravity load.

In this study, the apex connection design is demonstrated for a residential home package provided by an offsite construction company located in Alberta, Canada (11 m × 6.1 m Gabel roof footprint) with a slope of 8/12 and panel-A width of 1944 mm. Both gravity load case and wind load case were considered for a tributary area equal to the apex connection spacing. Assuming 600 mm c/c distance of the apex connections for a factored gravity load of 4 kPa, the axial force (C_x_) in the bar and internal hinge force (B_x_) at the apex location were calculated using Equations (2) and (3) and led to a solution of 5165 N and 1140 N respectively.

A 2-D finite element model was also developed utilizing commercially available general-purpose FE code, ABAQUS/CAE, distributed by SIMULIA Inc., Palo Alto, CA, USA. A two-node beam element (B31) was used to assemble the two LSL rafters with a cross-section of 140 mm × 38 mm and all the triangle apex connection components were made of steel C-channel section (30 mm × 40 mm × 40 mm) (Figure 17). It was assumed that the main folding bars connected by a pin in the actual connection setup act as one rigid link element. The three beams representing the apex connection, the secondary bar and the primary folding bar are connected by a hinge connector element (CONN3D2) (Points A, B and C in Figure 18). Then, the tie constraint was applied between the rafter and secondary bars of the apex connection. The tie constraint represents the screw connection between the rafter and the secondary bars, assuming that the number of screws provides sufficient rigidity to transfer the load from the rafter to the apex connection assembly. The same magnitude of gravity load was applied according to the previous analytical procedure and a comparison of the results shows that the FE model provides a reasonably close solution (axial force value in the main folding bar = 6570 N and apex hinge force at point B = 1879 N). Thus, the validation of this 2D FEM with the analytical model indicates that a more detailed modelling approach (e.g., contact simulation) can reveal the proper behaviour of this connection mechanism. However, this linear elastic analysis provides the basis to determine the internal forces and thereby an approximate diameter of the pins was obtained using Equation (5) [13].
(5)$$\mathrm{Shear}\;\mathrm{resistance}\;\mathrm{of}\;a\;\mathrm{pin},\;F_{v}=\frac{0.6\;A\;f_{up}}{\gamma_{M2}},$$
where d is the diameter of the pin, $f_{up}$ is the ultimate tensile strength of the pin and *A* is the cross-section area of a pin, and $\gamma_{M2}$ is the safety factor.

## 4. Screw Connection Requirement of Secondary Bars

In order to connect roof panels with the secondary bars of the proposed apex connection, commercially available screw (Simpson Strong-Tie SD screw), which is an alternative to common 10d nails, can be utilized. The screw nominal diameter and length of the screw are 4.5 mm and 38 mm, respectively. The in-plane component of load along the rafter plane must be resisted by the shear capacity of the screw connection between rafters and the secondary bars. Among all the load cases, gravity load has the highest magnitude. Thus, screw shear capacity was checked for the gravity load case only. Therefore, the number of holes in the secondary bar depends on the number of required screws. The analytical model for the nail and spike provided in CSA86-19 (Equation (6)) [15] was used to predict the unfactored shear capacity of the screw connection between the secondary bar and rafter. The predicted unfactored shear capacity per screw was 1898 N with the failure mode (e) of Equation (6). In this case study of a roof slope of 8/12, the secondary bar has a dimension of 410 mm between the pin holes (Figure 9) and seven holes are provided to install the screws. Thus, with seven screws, the total factored shear capacity is 10.63 kN. The screw shear force (summation of components of hinge forces of points B and C along the rafter in-plane) was obtained from the 2D analysis (Figure 18c). For the governing load (factored gravity load of 4 kPa), the resultant screw shear was 7.02 kN. Therefore, the number of screws provided for apex connection is safe in the case of in-plane shear resistance. Details of shear capacity analysis can be found in the Appendix B of this paper. The screw connection is also subjected to the highest withdrawal in case of wind load parallel to the ridge direction of the roof. According to the screw manufacturer’s technical data sheet, the design withdrawal capacity is 769.5 N per screw [16]. Additionally, the withdrawal force due to a wind withdrawal load is 2.1 kN. Thus, with seven screws, the withdrawal resistance is 5.4 kN. Further, the provided number of screws is sufficient to resist the hourly wind pressure of 0.85 kPa. Thus, providing seven holes in the secondary bar is sufficient to resist both the gravity and wind load in Edmonton, Alberta, Canada. The holes were provided at 50 mm c/c to avoid the splitting of the rafter of the roof panel. It is worthwhile to note that, to accommodate a higher load, the length of the secondary bar must be increased to provide proper screw spacing.
(6)$$N_{r}=\phi \;N_{u}n_{F\;}n_{s\;}$$
where:$N_{u}=n_{u}\left(K_{D}K_{SF}K_{T}\right)$ϕ=resistance factor for yielding failures=0.8$n_{u}=unit\;lateral\;resistance\;of\;screw$$n_{F\;}=number\;of\;screw\;in\;connection$$n_{s}=number\;of\;share\;planes\;per\;screw$

The unit lateral resistance of steel to timber screw connection per share plane is the smallest value calculated in accordance with the failure modes (a) to (f).
(7)$$\left(a\right)\;\;f_{1}d_{F}t_{1}$$
(8)$$\left(b\right)\;\;f_{2}d_{F}t_{2}$$
(9)$$\left(c\right)\;\;f_{1}d_{F}{}^{2}\left(\sqrt{\frac{1}{6}\frac{f_{3}}{\left(f_{1}+f_{3}\right)}\frac{f_{y}}{f_{1}}}+\frac{1}{5}\frac{t_{1}}{d_{F}}\right)$$
(10)$$\left(d\right)\;\;f_{1}d_{F}{}^{2}\left(\sqrt{\frac{1}{6}\frac{f_{3}}{\left(f_{1}+f_{3}\right)}\frac{f_{y}}{f_{1}}}+\frac{1}{5}\frac{t_{2}}{d_{F}}\right)$$
(11)$$\left(e\right)\;\;f_{1}d_{F}{}^{2}\frac{1}{5}\left(\frac{t_{1}}{d_{F}}+\frac{f_{2}}{f_{1}}\frac{t_{2}}{d_{F}}\right)$$
(12)$$\left(f\right)\;\;f_{1}d_{F}{}^{2}\sqrt{\frac{2}{3}\frac{f_{3}}{\left(f_{1}+f_{3}\right)}\frac{f_{y}}{f_{1}}}$$
where:$t_{1}=head-side\;member\;thickness\;\left(steel\;plate\;in\;this\;case\right)$$d_{F}=screw\;diamenter$$f_{2}=embedment\;strength\;of\;point\;side\;member\;\left(LSL\right)=50\;G\left(1-0.01d_{F}\right)$G=mean realtive density=0.5 for LSL$t_{2}=length\;of\;screw\;penetration$$f_{3}=embedment\;strength\;of\;point\;side\;member\;when\;failure\;in\;screw\;yielding=110G^{1.8}\left(1-0.01d_{F}\right)$$f_{y}=screw\;yield\;strength=50\left(16-d_{F}\right)$$f_{1}=embedment\;strength\;of\;steel\;plate=K_{sp}\left(\phi_{steel}/\phi_{wood}\right)f_{u}$$K_{sp}=2.7,\;\phi_{steel}=0.80,\;\phi_{wood}=0.8$

## 5. 3D Finite Element Modelling of Apex Connection

The actual folding apex connection is inherently three-dimensional in nature and involves complex interactions between the parts. The folding mechanism of apex connections technically represents, in principle, an extremely complex and highly indeterminate analytical problem with a wide range of geometrical nonlinearity and mechanical parameters affecting its behaviour to transfer the force and moment. These parameters include the rectangular slots (95 mm long) and U-shape slots (13 mm × 9 mm) in the primary folding link bar, V-shape cuts (14.5 mm × 7mm) in the Middle bar, contact between the main lock channel and primary link bars, contact between secondary lock channel and secondary link bars, pin bearing mechanism at the hole of main link bars and secondary bars, contact between the sidebar and main link bar (Figure 8, Figure 9, Figure 10, Figure 11, Figure 12 and Figure 13). Hence, three-dimensional elements were utilized in the FEM to understand the structural behaviour of this connection. Proper element selection for steel connection design is critically important. Abaqus provides several types of elements, such as continuum solid element, shell elements, membrane elements, rigid elements, and beam elements that can be used to simulate steel connections. The behaviour of these elements is characterized by five criteria, such as family, degrees of freedom, number of nodes, formulation, and integration. For example, the solid element library includes first-order (linear) interpolation elements and second-order (quadratic) interpolation elements in one, two, or three dimensions classifying as triangles and quadrilaterals for two dimensions; and tetrahedra, triangular prisms, and hexahedra (“bricks”) in three dimensions. Each class of element provides a choice for first-order (linear) interpolation elements and modified second-order interpolation elements in two or three dimensions. It is critical to select the correct element for a particular application to avoid hourglassing, shear and volumetric locking, overly stiff behaviour in bending and slow convergence with mesh refinement. It should be noted that the proposed apex connection may have large plastic deformations and high strain gradients in the pinhole regions of connection, as well as the presence of contact between the lock channel and link bars. As suggested in previous studies [17,18,19], first-order elements should be used to avoid mesh locking and convergence difficulties associated with contact, while modelling steel connection, 8-node linear brick (C3D8) element is preferable for the apex connection FEM. However, this element shows very high stiffness in bending; consequently, the incompatible mode of these elements (C3D8I) is implemented to improve their bending behaviour [20].

For any numerical modelling, the definition of material property is the most critical step that affects connection ductility and capacity. A detailed literature review of steel connections suggests assuming that the steel in this study is homogenous and isotropic with an elastic modulus of 200 GPa and Poisson’s ratio of 0.3 [19,21,22,23]. Since the variation of mechanical properties of steel is significantly low compared to other construction material, such as wood, in the absence of a coupon test, generic material properties specified in standards used for numerical simulation produces reasonably accurate results [24]. According to ASTM A1008, drawing quality steel sheets (DS) have yield strengths between 150 to 240 MPa [25]. While, for design purposes, selecting the nominal values is a conservative approach, the use of minimum nominal strength material property in numerical simulation underestimates the load-carrying capacity of the steel connection whereas adopting a maximum nominal strength property has the opposite effect [24]. Thus, to obtain proper connection behaviour the typical stress-strain relationship as depicted in Figure 19 [26] is adopted for the elements of all the parts fabricated using steel plate and A36 steel for all pin components (Figure 20) [24]. Since the main focus of the model is to understand the behaviour of the apex connection, the rafter material (LSL) was assumed to be isotropic with a Young’s modulus of 9000 MPa and Poisson’s ratio of 0.25.

In the structural analysis, it is common to make use of reduced or lower-dimensional element types with higher-dimensional element types in a single FE model, which is known as multiscale FEM. This approach is efficient and provides an improved solution to capture local structural features as well as global structural behaviour [27]. Figure 21 illustrates the complete mixed-dimensional FEM assembly of this study. To simplify the model, similar to the analytical approach, only two LSL rafters and the unfolded state of apex connection assembly were incorporated. However, all these components were modelled using solid 3D elements (8-node linear brick element). As can be observed from Figure 5, the apex connection has nine locations that require a hinge mechanism provided by steel pins. Despite well-established design rules and assembly procedures in American and European steel standards, numerous nonlinearities in the vicinity of the pinhole led to overly expensive calculations if fine-scale computation modelling is used. Consequently, to model this large assembly with a considerable number of pin joints, alternative computational strategies are a suitable option [28]. Abaqus provides the connector (CONN3D2) element to model any type of connection such as a hinge or a screw [20]. However, the connector element is a 1D element whereas the other components are 3D objects, so the number of degrees of freedom (DOFs) is not the same for all the objects. Hence, this multi-scale FE simulation requires a reasonable FE coupling method to blend mixed-dimensional finite elements at their interfaces to accomplish both displacement continuity and stress equilibrium. In this regard, the multipoint constraint (MPC) surface method is suitable for the static and dynamic analysis of linear or nonlinear structures and the interactions between the pin with the assembled elements can be modelled in an average sense with a more rigid way in ABAQUS [28]. The MPC method uses constraint equations for nodal displacements at the interface of mixed-dimensional elements. Thus, to model the hinge mechanism at all locations (as shown in Figure 22), reference points were generated on the center points of the connection and MPC-BEAM constraint was assigned to the hole surface with their corresponding reference point (RP). MPC-BEAM in ABAQUS uses a rigid interface method to connect nodes of different types of elements by creating rigid beams with respect to RP [20]. At each pin location connector element with a hinge, connection property was assigned to join corresponding parts. Tie constraint was applied at the interface between secondary bars and the LSL rafter assuming that the screw connection will act as a rigid joint. In the simulation of steel connection, the boundary condition is deemed to be significant and any inappropriate boundary conditions may cause completely different behaviour. Following the analytical model mentioned in the previous section, the lower two ends of the LSL rafter are assumed to be in the pinned (U_x_ = U_y_ = U_z_ = 0) support condition. In actual roof assembly, the OSB sheathing is nailed on the rafters, which provides stability against lateral buckling. To account for this lateral restraint, provided by the continuity of OSB sheathing panels, the rafter edges were assigned as the Z-symmetry coordinate system (U_z_ = UR_x_ = UR_y_ = 0), as shown in Figure 21.

Properly refined element mesh is also an influencing factor for any finite element simulation to obtain reliable results. Due to the presence of slots and pinholes, it was required to partition the complex geometry into several segments and assign an advancing front or medial axis meshing algorithm to generate elements with proper shape factors. For example, the U-shape slot in the main link bar in Figure 23 was partitioned by offsetting the half-circular face of the slot at every 1 mm and creating radial lines at every 9° angle. This technique generated a very refined mesh with a proper shape factor around the face of the model.

As mentioned previously, complex interactions exist between the surfaces of different parts of the apex connection to facilitate folding and self-locking mechanism, so this FE modelling requires contact simulations of different components to allow for a transmission of force from one part to another, specifically near the folding location where main lock channel and secondary lock channel provides a self-locking mechanism (Figure 24). In detail, a coulomb coefficient of friction equal to 0.3 is defined for sliding resistance in the surface-based contact approach. It should be noted that the interface of the contact surface must be close and the penalty technique enforcement was used in contact enforcement since this approach has more flexibility and recommended method in steel connection modelling [19].

## 6. Results

Four governing loading combinations from the Canadian Building Code were considered for this numerical analysis. These load combinations account for all the combinations, including gravity load (dead, live and snow) and lateral load (wind load Figure 25), that will lead to maximum effects for both strength and serviceability. The design loads applied to the roof structure were assigned based on the National Building Code of Canada and the building location in Alberta, Canada. These were specified at a snow load of 2.25 kPa, other non-structural components account for another 0.5 kPa of dead load and hourly wind pressure of 0.85 kPa. Thus, the total factored gravity load was 4 kPa. Wind loads were calculated using the static procedure of NBCC and based on the gust effect and pressure coefficient (as illustrated in Figure 25) for both wind perpendicular to ridge and wind parallel to ridge direction. For both the gravity load and lateral load cases, the partial loading scenario was also checked following the NBCC. The uniformly distributed load was applied on the LSL assuming apex connection spacing of 600 mm c/c since the maximum spacing of the roof panel rafter must not exceed the corresponding value to take advantage of the load sharing system effect.

Load case a: 1.25D + 1.5S +1.0L

Load case b: 0.9D + 1.4W + 0.5S

Load case c: 0.9D + 1.5S + 0.4W

Load case d: 0.9D + 1.4W

where D = dead load, L = live load, S = snow load and W = wind load.

**Figure 25 materials-15-07457-f025:**
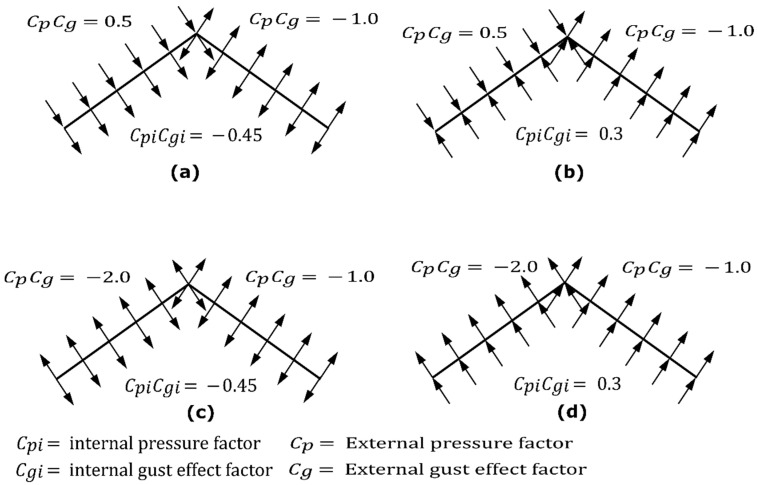
External and internal gust and pressure factors for wind load cases (**a**,**b**) for wind perpendicular to ridge and (**c**,**d**) for wind parallel to ridge.

Figure 26 and Figure 27 show the deflected shape of the assembly under gravity load and von Mises stress of the apex connections parts. Table 1 summarizes the resultant von Mises stress and PEEQ value observed in different load cases. As the link assembly is subjected to both bending and compression for the gravity load, the maximum von Mises stress was observed around the U-slot of the primary folding link bar (Figure 28 and Figure 29). It was observed that the load had the largest impact on the stress distribution in the proximity of the U-slot of the primary folding link bar. Figure 29 depicts the stress distribution near the U-slot indicating the highest stress concentration zone and probability of fracture initiation zone at ultimate failure load. A comparison of the stress contour plotted in Figure 30 shows that all other parts including the main lock channel, middle bar, secondary lock channel, sidebars and secondary folding link bars are within the elastic limit. As expected, among these components, maximum von Mises stress was observed in the contact zone of the main lock channel and secondary lock channel which confirms the effectiveness of the locking mechanism of the assembly. Maximum von Mises stress in the main lock channel and secondary lock channel were 55.2 MPa and 51.2 MPa respectively, whereas the ultimate strength of AISI 1008 is 353.3 MPa. Thus, the channels are well below the ultimate strength of the material for a factored gravity load of 4 kPa.

In order to obtain the connection capacity, a benchmark for the failure mechanism and failure criteria for any numerical analysis are required. In the case of the experimental investigation of any connection, the failure point of an assembly can be distinct by observing a situation when the assembly exhibits a substantial loss in load-carrying capacity or the presence of a rupture mechanism. However numerical models continue to obtain results until it fails to converge on a solution which can be different from the actual failure state. Thus, it is important to establish proper failure criteria for the numerical modelling approach. The literature review has revealed that, in the case of steel connection, two types of failure criteria are considered, namely (a) strength criterion and (b) deformation criterion. For example, experimental studies on the bearing resistance of a connection adopted a design equation based on the maximum loads from tests in References [29,30,31,32], even though significantly larger deformation in the specimen was observed. In contrast, according to the later criterion, failure is considered to be the applied load measured at a specific acceptable deformation level depending on the application [33]. For instance, the bearing failure study of cold-formed steel bolted connections by Salih et al. [34] adopted a 3 mm extension limit. Additionally, Eurocode 3 design provisions for steel connections are based on a 3.0 mm deformation limit under ultimate conditions which ensures the deformation under service loads to be 1.0 mm [33]. Thus, for design purposes, deformation-based criteria are more appropriate. Hence, the equivalent plastic strain of a material (PEEQ) in Abaqus, was adopted to implement as a design capacity criterion for the cross-section of the parts. PEEQ is a scalar measurement that is used to represent the material’s inelastic deformation and if this variable is greater than zero, the material has yielded. Thus, PEEQ indicates the local ductility and fracture tendency of steel members [35]. In the case of classical (Mises) plasticity, PEEQ is obtained by using the following equation:(13)$$PEEQ=\bar{\varepsilon_{p}}{|}_{0}+\int_{0}^{t}\sqrt{\frac{2}{3}\dot{\varepsilon_{p}}:\;\dot{\varepsilon_{p}}}dt$$
where $\bar{\varepsilon_{p}}{|}_{0}\;$ is the initial equivalent plastic strain and $\dot{\;\varepsilon_{p}}\;\;\mathrm{is}\;\mathrm{the}\;\mathrm{tensorial}\;\mathrm{form}\;\mathrm{of}\;\mathrm{plastic}\;\mathrm{strain}\;\mathrm{rate}$.

In order to investigate the performance of the connection in service, the following hypothesis was adopted:

“The apex connection components must be sized so that all materials remain in the elastic range and their elastic deformations have negligible values. This ensures the deformation of the connection will be returned to its original state after the load is removed.”

As can be observed from the deformation and stress distribution of the primary folding link bar, the possible mechanism of failure is fracture propagation near the U-slot. Thus, the design requires avoiding any form of localized plastic deformation near this zone. If the PEEQ value is zero, then it can be concluded that the assembly is in the elastic range under design load. Figure 30, Figure 31, Figure 32 and Figure 33 illustrate the von Misses stress and PEEQ plot of the apex connection assembly in the most critical load cases. The PEEQ value analysis of all the critical load cases shows non-zero plastic strain for load case-a only (Figure 34). The PEEQ value for a factored gravity load of 4 kPa and the cumulative plastic deformation is concentrated in the middle location of the U-slot; 12 mesh (C3D8I) elements (approximately 64 mm^3^ volume) near the U-slot of the primary folding link bar have PEEQ value greater than zero with a maximum value of 1.7810 × 10^−3^. As can be observed from Figure 35, the computed strain levels in those 12 elements have exceeded the defined yield value and are in the strain hardening stage. Consequently, there will be a 0.185 mm permanent deformation near the U-slot zone of the one-side main folding link bar. Therefore, the main folding link has the probability of a total of 0.4 mm shortening for a specified gravity load of 4 kPa. This will cause tension force at the secondary bar, ultimately increasing the uplift force on the fastener used to connect the bar with the roof panels. Additionally, the relatively long winter in Canada poses a risk of fatigue and residual stress on the connection, so a PEEQ value equal to zero will be a safe option to ensure that the assembly components remain in the elastic zone. Therefore, it can be concluded that a revised cross-section is required to support a 4 kPa gravity load. However, at a factored load of 2.6 kPa, the cross-section shown in Figure 8 with PEEQ values equal to zero was observed. Thus, the primary folding link bar of the cross-section shown in Figure 8 can be used for a factored gravity load of 2.6 kPa. An enhanced cross-section (Figure 36) is required to support a factored gravity load of 4 kPa, which was obtained by running the numerical model with various trial cross-section sizes until the failure criterion was met. It is worthwhile to note that the initial primary folding link bar (as shown in Figure 8) has a cross-section of 30 mm × 38 mm, whereas the revised cross-section is 30 mm × 50 mm (as illustrated in Figure 36). All other elements, such as the pinhole and u-slot, are the same as initially designed (Figure 8).

To observe the load-deflection behaviour of the apex connection assembly, a gravity load up to 12.7 kPa was applied since the load case-a is the governing case. Figure 37 illustrates the mid-point deflection vs. load plot of the primary folding link bars assembly, which ensures reasonably low deflection of the overall assembly at the factored gravity load of 4 kPa.

The design of the connection also requires checking the pin shear and bearing capacity of the steel sheet near the hole. Thus, another numerical model was developed to simulate a uniaxial tension test on the pin joint in the middle of the main link bar to observe the localized effect due to the interaction of the pin and the holes of the main link bar (Figure 38). In this case, the pin was modelled using a 3D solid element (C3D8I). Hard contact was defined between the pinhole of the folding bars and the pin. Figure 39 illustrates the von Mises stress distribution and PEEQ value due to the maximum applied load of 15 kN in this case. Using EN 1993-1-8 [13] (Equation (5)) the predicted pin shear capacity was 8919 N, whereas the numerical model shows the shear capacity to be 8070 N, considering no plastic strain at any location of the pin (PEEQ = 0) (Figure 40). However, the literature review has shown that, in experimental investigation, connection capacity is defined based on certain deformation levels [29,30,31,32,33,34]. In the absence of the experimental test, it is a conservative design approach to adopt the PEEQ value equal to zero as the capacity benchmark for the pin. The maximum stress in the pin was 366.8 MPa (for PEEQ = 0 condition) and the ultimate stress for A-36 steel is 586.7 Mpa. However, for a factored gravity load of 4 kPa (governing load for axial force), the axial force value in the main folding link bar was 5898 N. The maximum stress of the pin for this axial force was 286.9 Mpa. Therefore, the pin section is safe in the case of the factored gravity load of 4 kPa and hourly wind pressure of 0.85 kPa.

For the pin connection, another important mode of failure is bearing for the steel plate and pin. It is required to have a benchmark to understand the bearing failure mechanism. Details of the bearing failure mechanism can be found in single-lap bolt connection studies. Ideally, bolts in a single-lap connection and the pin connection have similar structural behaviour such as shear failure, bearing failure and net section failure, the only difference is that the bolt connection has resistance due to clamping friction, whereas the pin provides free rotation. As the load is gradually applied to the bolted connection, the major force transfer would be friction between the contact surfaces. Once the applied force exceeds the friction capacity of the connected members slip relative to each other until they bear on the bolts, contact with the hole interaction dominates the connection performance similar to the pin connection. Thus, the bearing failure benchmark can be obtained from the bolted connection bearing stress review. The literature review has shown that, in the case of bearing failure, a 3 mm hole elongation level is considered to be the ultimate capacity of the connection, thereby ensuring 1 mm deformation at the serviceability limit state [33]. Using EN 1993-1-8 (Equation (8)) [13], the bearing resistance capacity for the plate and pin was calculated to be 12,527 N, however, at this load level, the numerical model shows hole elongation of 0.56 mm which Is below the threshold limit for bearing failure criterion for serviceability limit state set by Eurocode. Additionally, the maximum design axial force (5898 N) for a factored gravity load of 4 kPa is less than the pin shear capacity (8070 N). Maximum bearing stress in the pinhole was 180.7 Mpa (for PEEQ = 0) and the ultimate strength of ASIS 1008 steel was 353.4 Mpa. Therefore, the assembly is safe in case of the factored gravity load of 4 kPa and wind pressure of 0.85 kPa in bearing. Thus, it can be concluded that the design capacity of the connection assembly is governed by the strain level near the U-slot of the main link bar.
(14)$$\mathrm{Bearing}\;\mathrm{resistance}\;\mathrm{of}\;\mathrm{the}\;\mathrm{plate}\;\mathrm{and}\;\mathrm{the}\;\mathrm{pin},\;F_{b}=\frac{1.5\;t\;d\;f_{y}}{\gamma_{M0}},$$
where *d* is the diameter of the pin, $f_{y}$ the lower the yield strengths of the pin and the connected part, *t* is the thickness of the plate, and $\gamma_{M0}$ is the safety factor.

## 7. Conclusions

A novel apex connection concept was presented in this paper. The 3D printout of the assembly demonstrates connection effectiveness in providing the folding, unfolding and self-locking mechanism. The folding mechanism of the connection provides easy transportation of the roof panel assembly, and the self-locking mechanism reduces the onsite workload. The connection can be installed at the offsite facility. In order to reduce the onsite workload and enhance productivity, easy installation of the intercomponent connections is required. In this respect, the folding design of the apex connection will facilitate the erection of two panels simultaneously and reduces the crane lifting number. Furthermore, the self-locking mechanism removes the onsite installation activity of the apex connection. Consequently, this novel apex connection is expected to improve onsite productivity. However, the comparison with current truss base roof fabrication requires future time study for this panelized roof system with this folding apex connection.

The validation of 2D FEM with the analytical solution confirms the adequacy of the model for the connection force analysis. The 3D FEM results of PEEQ show that two different C-sections (30 mm × 38 mm and 30 mm × 50 mm) of the main folding bar are required for specified gravity loads of 2.6 kPa and 4 kPa, respectively. The shear capacity of the pin (d = 6.35 mm) obtained from the analytical model was higher than the predicted capacity of FEM since the former procedure is developed based on certain deformation levels in the experimental investigation. To obtain the ultimate strength of a steel connection, validation of the numerical model with an experimental test is necessary. This study was limited by the lack of material coupon tests and actual assembly tests. The benchmark of the PEEQ value equal to zero in determining the design capacity of the assembly is a conservative approach. Therefore, to establish the proper ultimate capacity and serviceability limit of the present connection, actual testing of the connection is recommended. However, the obtained cross-section from this study provides the basis for the fabrication and testing of the assembly in future.

As part of a future study, long-term performance, such as fatigue and durability of this connection, must be investigated. Following the industry practice of light frame connectors such as wood I-Joist hangers, the fabrication of components of the apex connection must use galvanized sheet metal to enhance weatherability.

## Figures and Tables

**Figure 1 materials-15-07457-f001:**
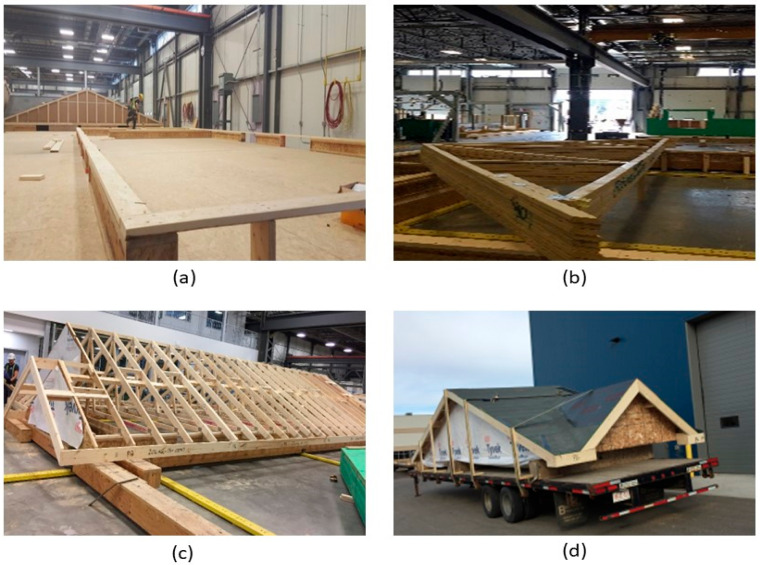
Roof production (**a**) setup jig, (**b**) unloaded truss, (**c**) attaching roof components, and (**d**) small roof module on the transportation trailer (courtesy of ACQBUILT Inc., Edmonton, AB, Canada) [6].

**Figure 2 materials-15-07457-f002:**
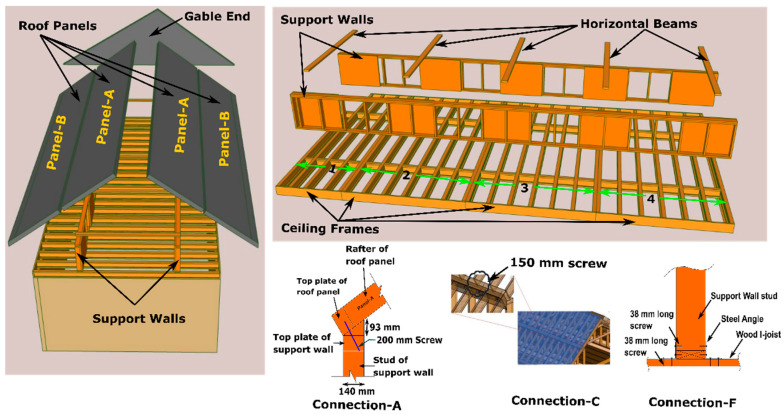
Light wood frame panelized roof concept.

**Figure 3 materials-15-07457-f003:**
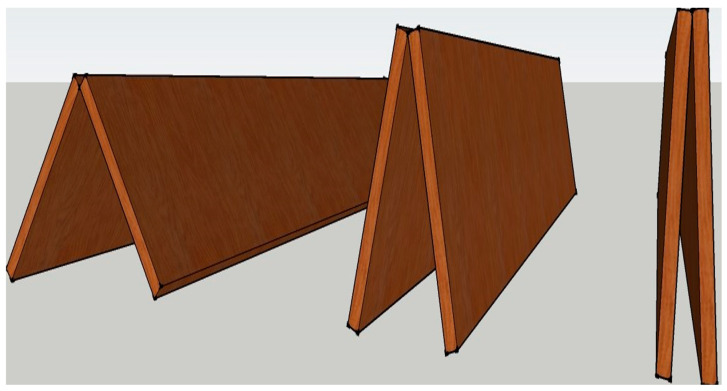
Roof panel folding.

**Figure 6 materials-15-07457-f006:**
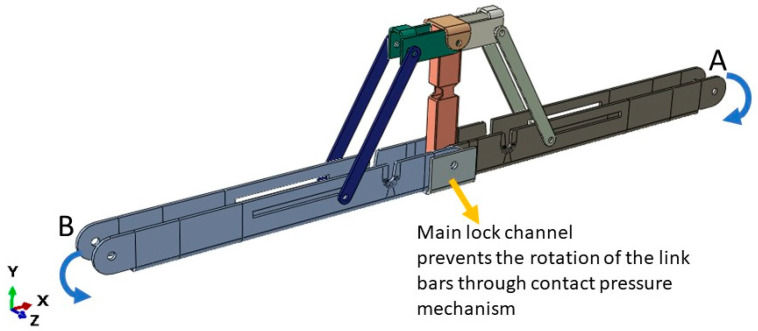
The main lock channel resisting clockwise rotation at point A.

**Figure 7 materials-15-07457-f007:**
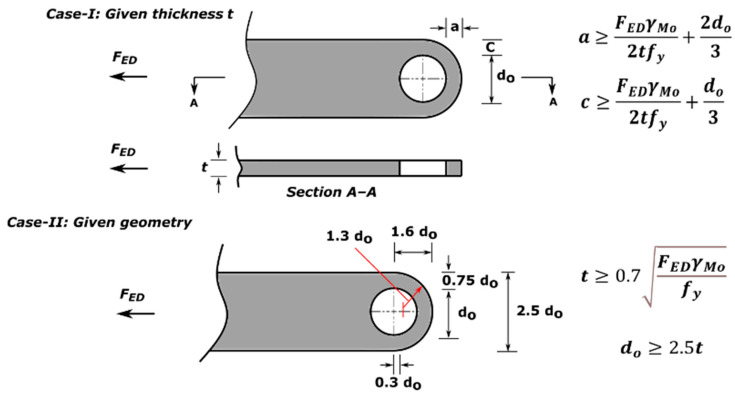
Edge distance and end distance requirement for pin connection according to EN 1993-1-8 [13].

**Figure 8 materials-15-07457-f008:**
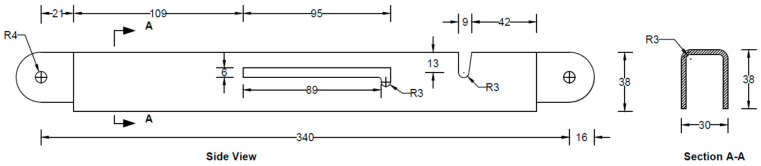
Details of primary folding link bars (all dimensions are in mm, steel plate gauge 11).

**Figure 9 materials-15-07457-f009:**
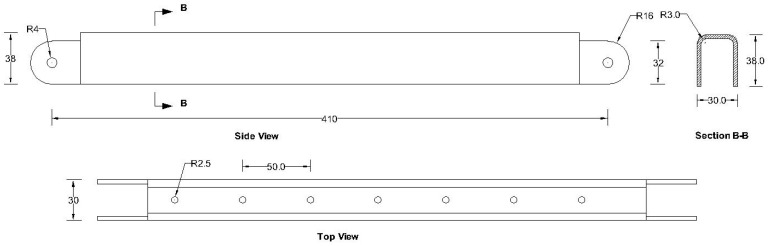
Details of Secondary bars (all dimensions are in mm, steel plate gauge 11).

**Figure 10 materials-15-07457-f010:**
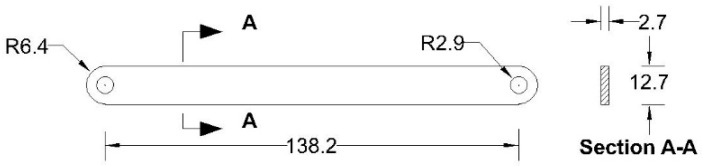
Details of Sidebars (all dimensions are in mm, steel plate gauge 12).

**Figure 11 materials-15-07457-f011:**
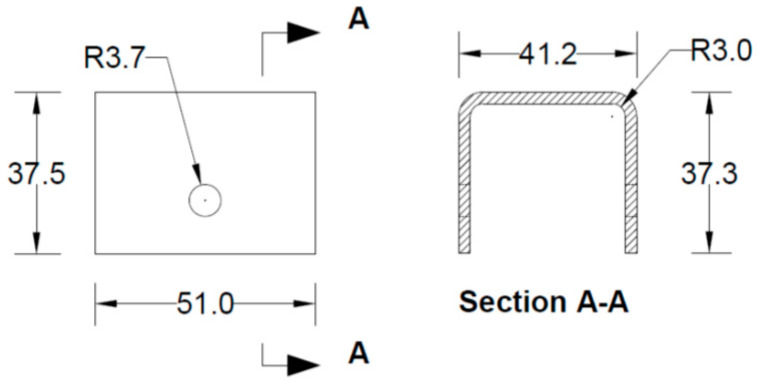
Details of main lock channel (all dimensions are in mm, steel plate gauge 11).

**Figure 12 materials-15-07457-f012:**
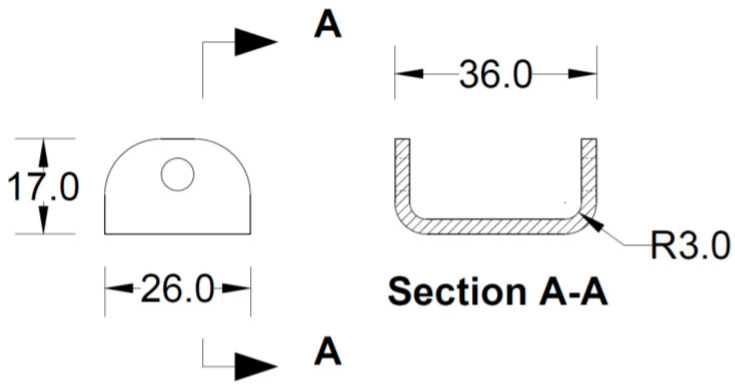
Details of secondary lock channel (all dimensions are in mm, steel plate gauge 12).

**Figure 13 materials-15-07457-f013:**
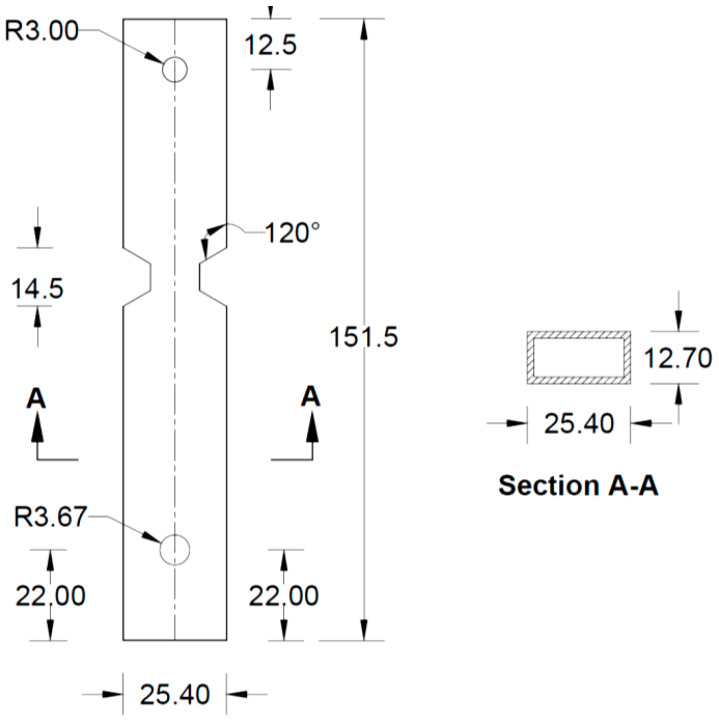
Details of Middle bar (all dimensions are in mm, HSS section).

**Figure 14 materials-15-07457-f014:**
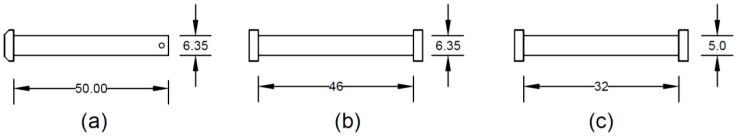
(**a**) Pin for connecting secondary bars at the apex point; (**b**) pin at the middle of the main link bar; and (**c**) Pin for sidebars.

**Figure 17 materials-15-07457-f017:**
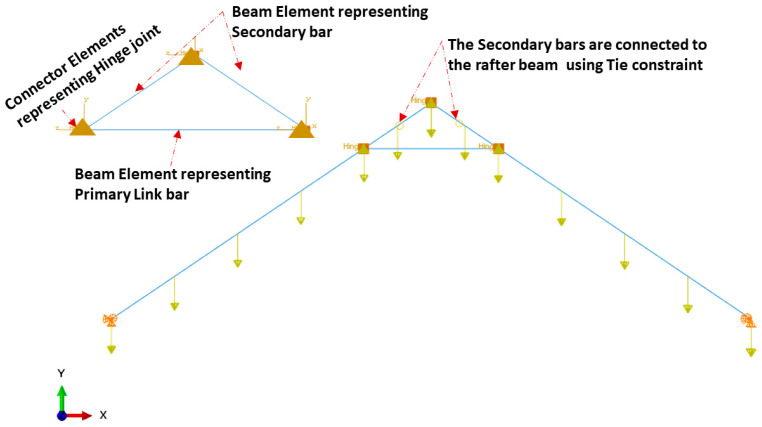
Loading and boundary condition 2-D numerical model of the apex connection.

**Figure 18 materials-15-07457-f018:**
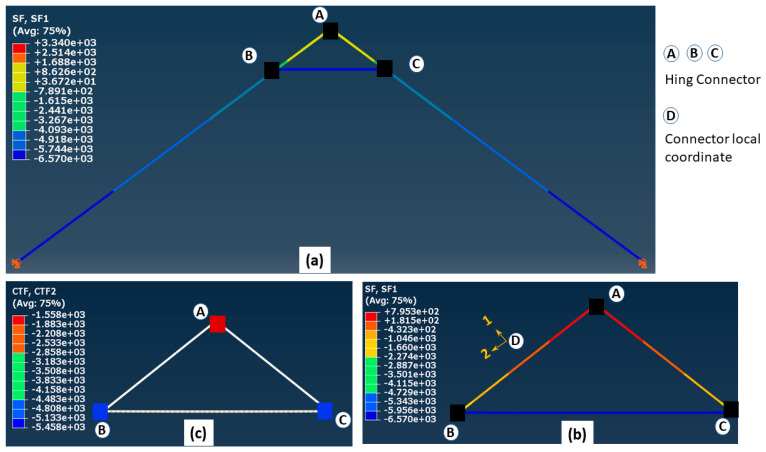
(**a**) Axial forces in the 2D assembly, (**b**) axial force in the apex connection, and (**c**) forces in the pins of apex connection (unit in the figure is N).

**Figure 19 materials-15-07457-f019:**
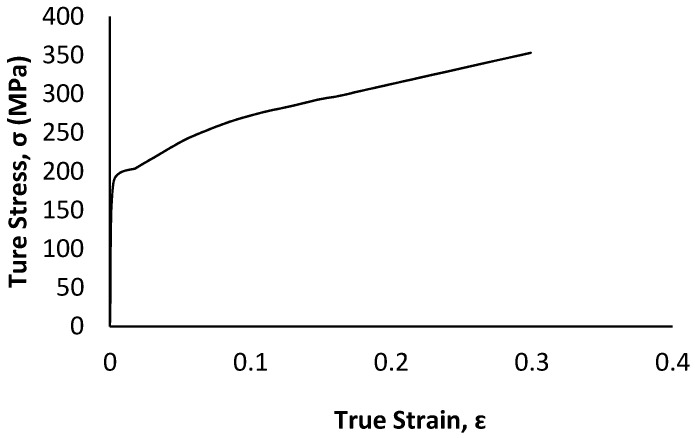
The true stress-true strain curves of an AISI 1008 steel (Data obtained with permission from [26], 2014, Elsevier).

**Figure 20 materials-15-07457-f020:**
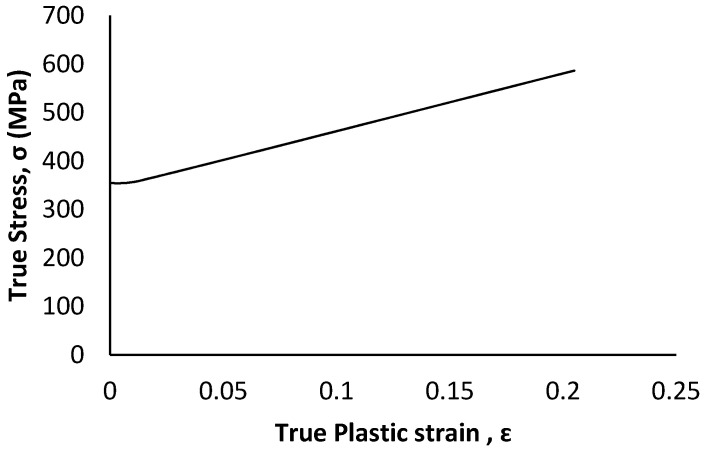
The true stress-true plastic strain curves of A36 steel [24].

**Figure 21 materials-15-07457-f021:**
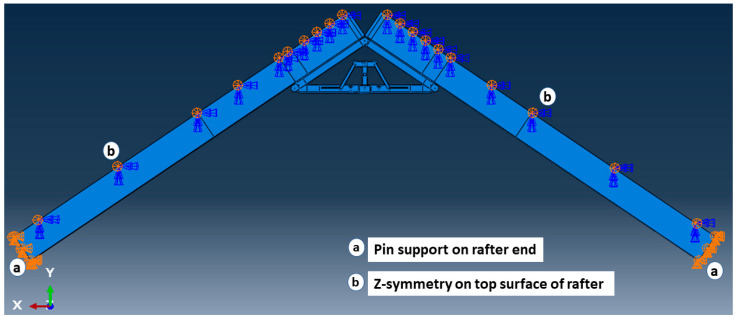
3D assembly model and boundary condition.

**Figure 22 materials-15-07457-f022:**
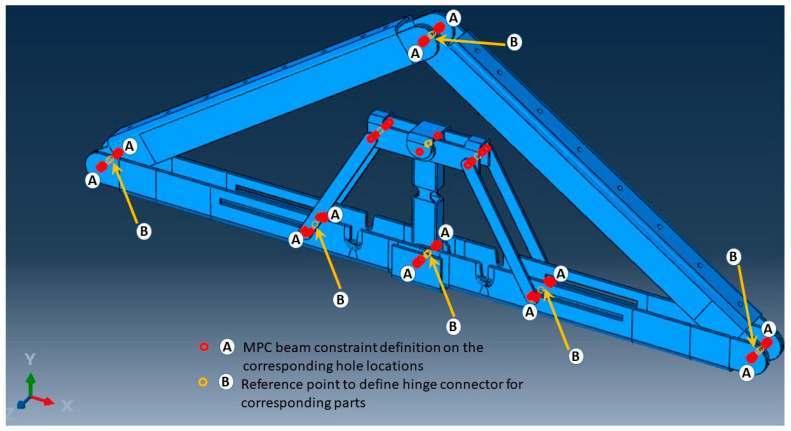
MPC beam constraint location for the hinge connection.

**Figure 23 materials-15-07457-f023:**
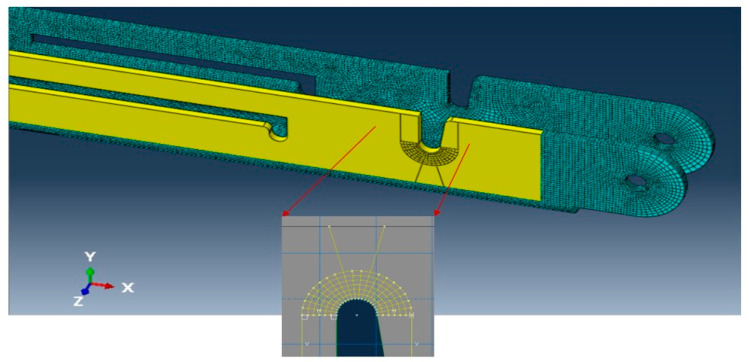
Advanced meshing application.

**Figure 24 materials-15-07457-f024:**
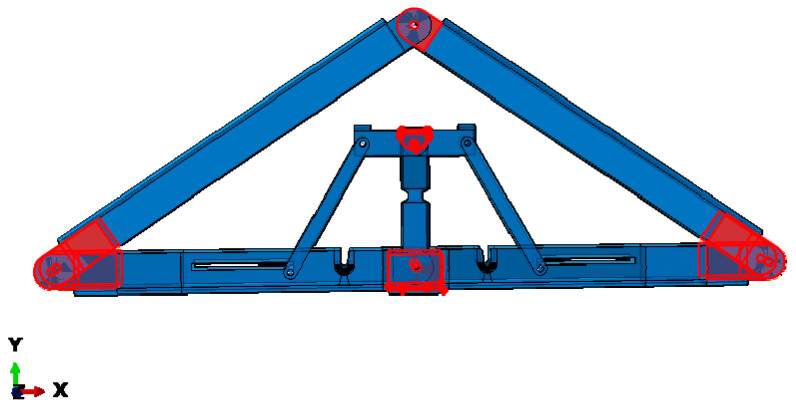
Contact surface interaction locations.

**Figure 26 materials-15-07457-f026:**
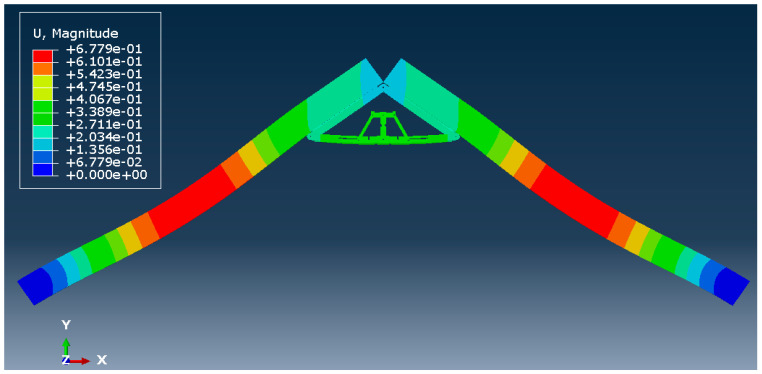
The deflected shape of the structure for a factored gravity load of 4 kPa.

**Figure 27 materials-15-07457-f027:**
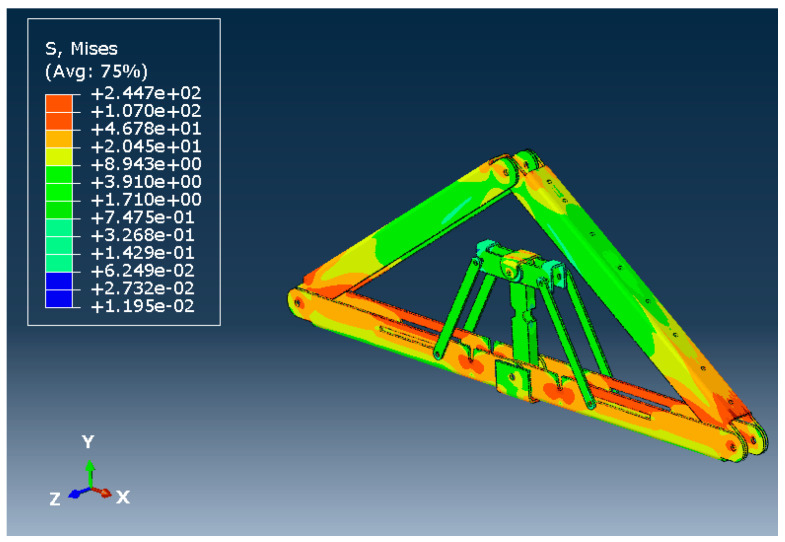
Von Misses stress (MPa) of the Apex connection (factored gravity load of 4 kPa).

**Figure 28 materials-15-07457-f028:**
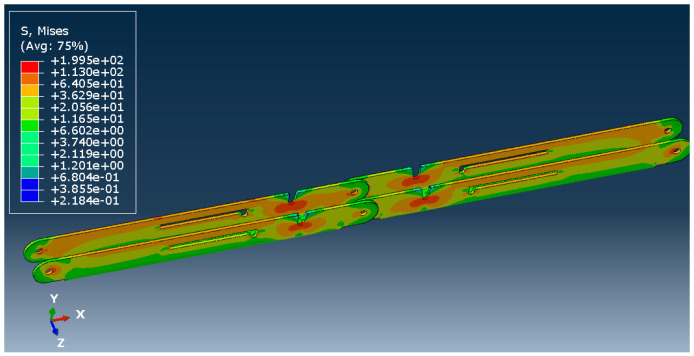
Von Mises Stress (MPa) of primary folding link bars (factored gravity load of 4 kPa).

**Figure 29 materials-15-07457-f029:**
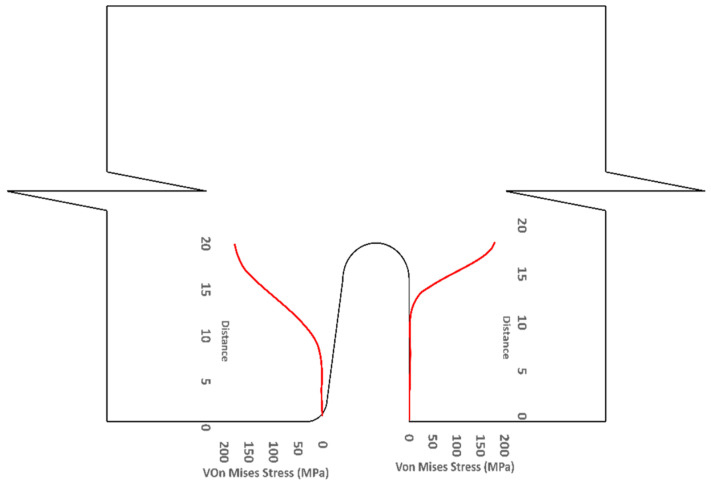
Stress (MPa) distribution near (red line) the vicinity of the U-slot of primary folding link bar.

**Figure 30 materials-15-07457-f030:**
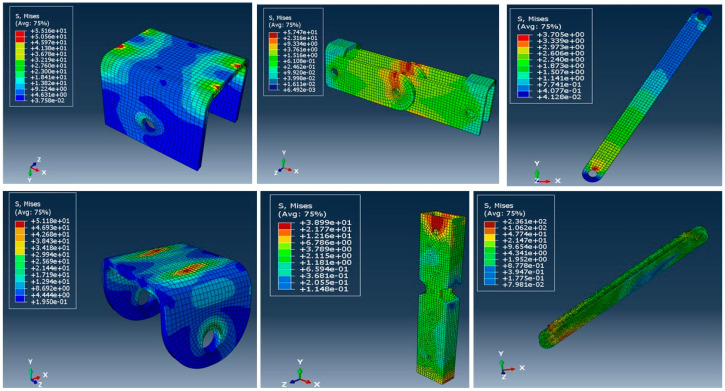
Stress (MPa) contour plot of different parts for a factored gravity load of 4 kPa.

**Figure 31 materials-15-07457-f031:**
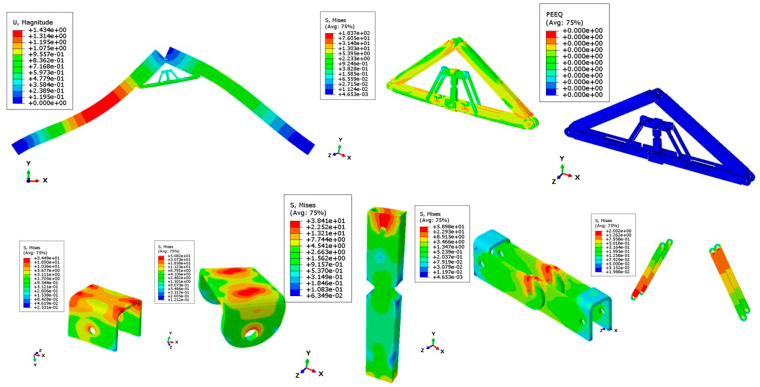
Stress (MPa) in wind load case (100% load removed from one side for hourly wind pressure of 0.85 kPa).

**Figure 32 materials-15-07457-f032:**
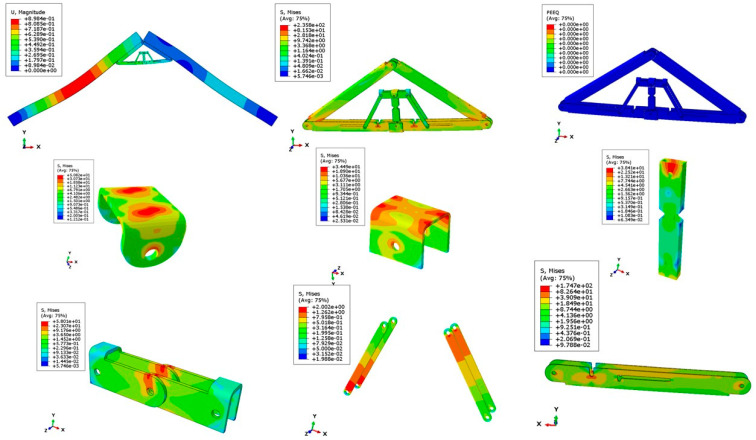
Stress (MPa) in different parts for partial snow load case (specified snow load of 2.25 kPa).

**Figure 33 materials-15-07457-f033:**
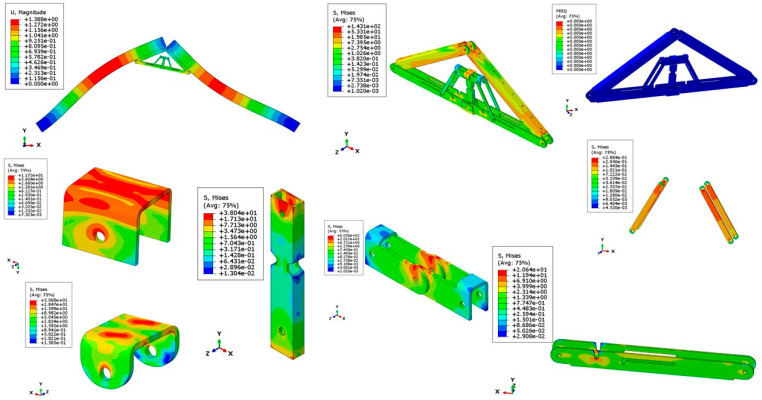
Stress (MPa) for partial wind load case uplift (hourly wind pressure= 0.85 kPa).

**Figure 34 materials-15-07457-f034:**
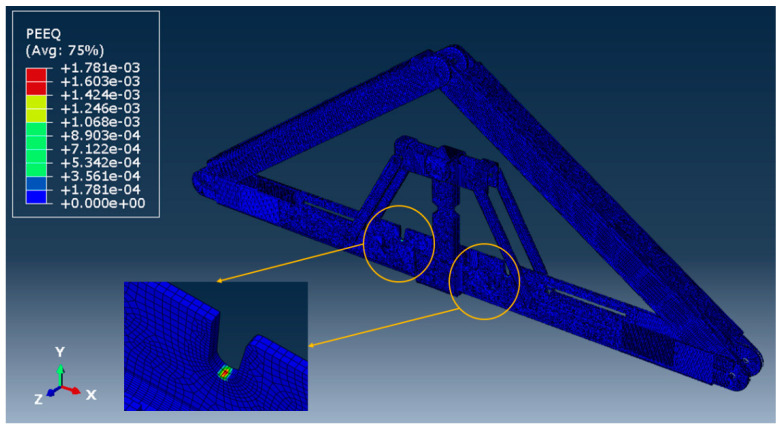
PEEQ value for the link assembly for a factored gravity loading of 4 kPa.

**Figure 35 materials-15-07457-f035:**
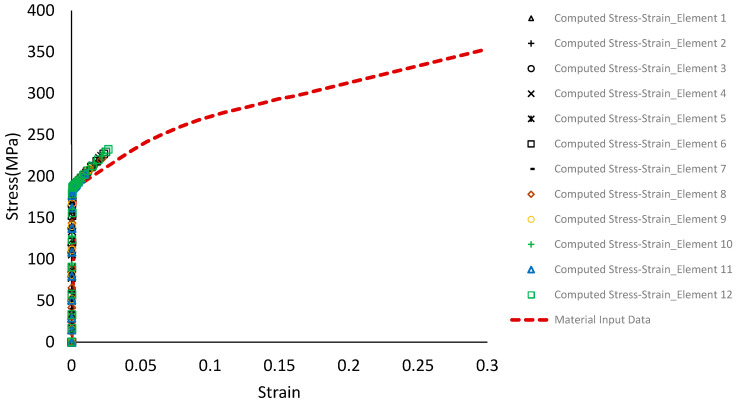
Stress-Strain near the U-slot of primary folding link bar.

**Figure 36 materials-15-07457-f036:**
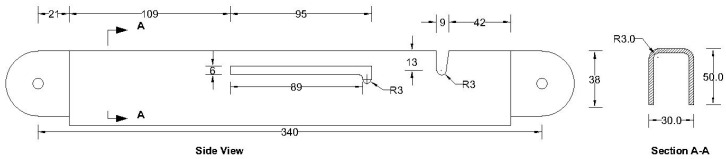
Revised primary folding link bar cross-section.

**Figure 37 materials-15-07457-f037:**
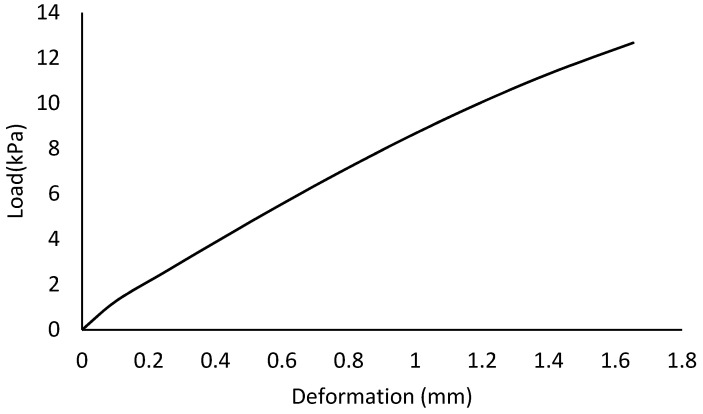
Deflection at the midpoint of the primary folding link bar assembly.

**Figure 38 materials-15-07457-f038:**
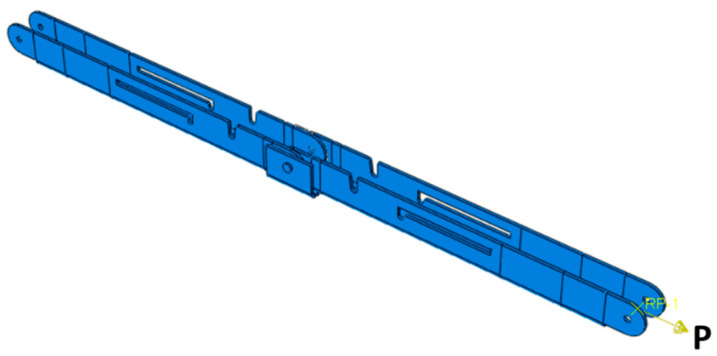
Uniaxial tension simulation of primary folding link bars for bearing and pin shear capacity.

**Figure 39 materials-15-07457-f039:**
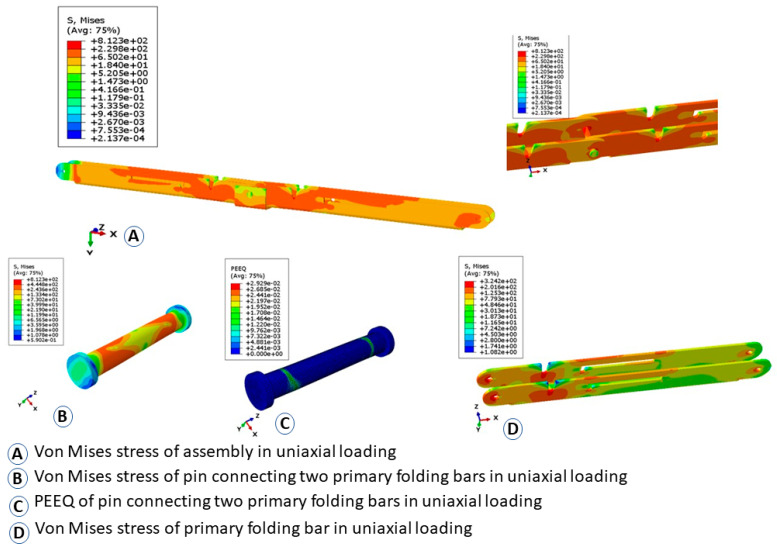
Von Mises stress (Mpa) and PEEQ plot of uniaxial loading of primary folding link bar and pin (P = 15,000 N).

**Figure 40 materials-15-07457-f040:**
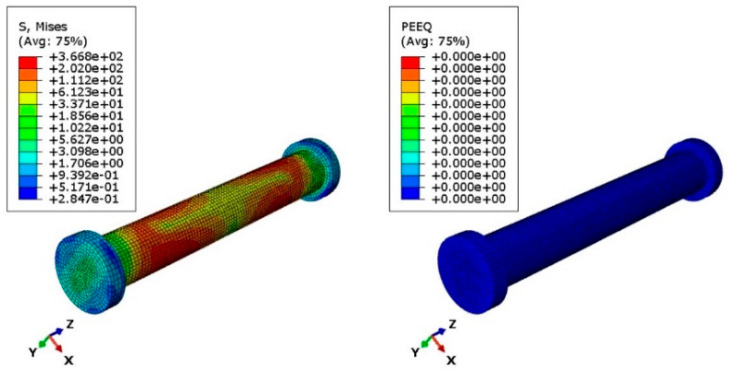
Von Mises stress (Mpa) and PEEQ plot of the pin at the predicted design capacity using the EN 1993-1-8 equation (P = 8070 N).

**Table 1 materials-15-07457-t001:** Summary of results in different load cases.

Part Name	Load Case	Maximum Permissible Plastic Stain (PEEQ) (mm/mm)	Observed Maximum Plastic Strain (PEEQ) (mm/mm)	Maximum Permissible Von Mises Stress (MPa)	Observed Maximum Stress (MPa)
Primary folding link bar (30 mm × 38 mm c-section in Figure 8)	^1^ Load case a	0	$1.7810\times 10^{-3}$	186.7	199.50
	^2^ Load case b		0		81.03
	^2^ Load case c		$3.7910\times 10^{-5}$		184.66
	^3^ Load case d		0		177.56
Secondary bar	^1^ Load case a	0	0	186.7	117.61
	^2^ Load case b		0		112.95
	^2^ Load case c		0		116.72
	^3^ Load case d		0		186.76
Main lock channel	^1^ Load case a	0	0	186.7	55.10
	^2^ Load case b		0		33.59
	^2^ Load case c		0		60.58
	^3^ Load case d		0		47.75
Secondary lock channel	^1^ Load case a	0	0	186.7	51.18
	^2^ Load case b		0		50.79
	^2^ Load case c		0		51.24
	^3^ Load case d		0		50.84
Secondary Folding link bars	^1^ Load case a	0	0	186.7	57.47
	^2^ Load case b		0		59.18
	^2^ Load case c		0		57.26
	^3^ Load case d		0		62.61
Side bars	^1^ Load case a	0	0	186.7	3.70
	^2^ Load case b		0		1.72
	^2^ Load case c		0		3.81
	Load case d		0		2.48
Middle bar	^1^ Load case a	0		186.7	38.99
	^2^ Load case b		0		38.34
	^2^ Load case c		0		39.09
	^3^ Load case d		0		40.34

Note: ^1^ Gravity load case, total load = 4 kPa; ^2,3^ For the load cases wind pressure 0.85 kPa and specified snow load 2.25 kPa.

## Data Availability

Not applicable.

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
