# Peer review of "Novel Apex Connection for Light Wood Frame Panelized Roof"

_materials, 2022, doi:10.3390/ma15217457_

Round 1
Reviewer 1 Report
Review: Manuscript Materials – 1909149
Title: Novel apex connection for light wood frame panelized roof
The article presents a new concept of apex connection for wooden roof panels. The idea is very interesting, but the article needs major improvements. In general, it is an interesting article, with great applicability and based on the numerical analysis of the structure. However, it requires a more logical and rigorous organization of information, some of which can be summarized in tables (the loads applied to the elements, material characteristics, types of finished elements used, etc.)
1. The introduction is poor in terms of wooden roof systems and wood-based materials used, both as research addressed and as figures.
2. The mention of Figure 2 in the text (page 1, paragraph 46) is not made by simply copying the title of the figure. It would be preferable to explain the component elements with reference to the respective figures. The title of figure 2 is Innovative panelized roof concept - what is the innovative concept in this figure?
3. Figure 16 requires more readability and higher quality
4. On page 11, the expressions stated from a) - f) should be numbered as equations, according to the template.
5. Figure 24 may be missing
6. The results must be summarized in a table, in which you present the stresses, displacements, admissible values and possibly some observations.
Finally, I believe that the article can be published after the recommended major revisions.
Author Response
Please find the response attached.

Reviewer 2 Report
This paper presents an apex connection for light frame panelized roof. A 3-D finite element model is developed. The authors of the paper carried out 3D finite element modelling. Detailed discussions about the triangular hinge apex connection are made. Results from the modeling have merits for application to light frame buildings. The paper is logic and organized. Therefore, I recommend that this paper is accepted for publication.
Author Response
Thank you for your comments and suggestions

Reviewer 3 Report
In this paper, a new concept of vertex connection of wood structure is proposed, and the strength of the connection point is tested by 3D printing and finite element analysis. However, there are several questions not mentioned or clearly clarified by the authors, so the manuscript should be revised before publication.
1) In line 56, Page 3,the first indent should be set .
2) Inline 202, page 9, The text in the picture is not displayed clearly.
3) Durability and weatherability of untested components.
4) In 630 lines, 28 pages, file format error.
5) Differences from previous component structures not specified.
6) It is not clear how much performance improvement the component has over the traditional or other component.
7) On page 10, line 204, the picture in Figure 17 (a) (b) (c) is blurred and should be modified.
8) On page 14, line 334, the indication line in Figure 20 is not clear and should be modified.
9) On page 14, line 336, Figure 20 indicates that the ghost is not clear and should be modified.
On page 21, line 485, the simulation diagram in Figure 33 is not highly recognizable to the background, so it should be modified.
Author Response
Please find the response attacehd

Round 2
Reviewer 1 Report
The paper has been improved considerably. It can be published after making the minor changes specified below:
Figure 19 - correct the unit of measure for deformation, respectively for stress. Deformation is not measured in MPa.
Figure 35 - correct to the unit of measure MPa (the letter P must be capitalized).
Author Response
Figure 19 was updated to correct the units in the graph in the revised manuscript
Figure 35 was modified to correct the unit of stress in the revised manuscript